# Worldwide Late Pleistocene and Early Holocene population declines in extant megafauna are associated with *Homo sapiens* expansion rather than climate change

Juraj Bergman [1,2] ✉, Rasmus Ø. Pedersen [1,2], Erick J. Lundgren [1,2,3], Rhys T. Lemoine[1,2], Sophie Monsarrat [1,2,4], Elena A. Pearce[1,2], Mikkel H. Schierup [5] & Jens-Christian Svenning [1,2]

The worldwide extinction of megafauna during the Late Pleistocene and Early Holocene is evident from the fossil record, with dominant theories suggesting a climate, human or combined impact cause. Consequently, two disparate scenarios are possible for the surviving megafauna during this time period - they could have declined due to similar pressures, or increased in population size due to reductions in competition or other biotic pressures. We therefore infer population histories of 139 extant megafauna species using genomic data which reveal population declines in 91% of species throughout the Quaternary period, with larger species experiencing the strongest decreases. Declines become ubiquitous 32–76 kya across all landmasses, a pattern better explained by worldwide *Homo sapiens* expansion than by changes in climate. We estimate that, in consequence, total megafauna abundance, biomass, and energy turnover decreased by 92–95% over the past 50,000 years, implying major human-driven ecosystem restructuring at a global scale.

The late-Quaternary extinction event[1,2] is characterised by the selective extinction of large-bodied animals (megafauna) at a global scale. At the present date, only a small fraction of this prehistorically speciose group[2–5] persists in rapidly diminishing communities, many of which face an immediate threat of extinction[6,7]. The causes of megafauna decline have been subject to long-standing debate, with fluctuations in paleoclimate and the spread of *Homo sapiens* emerging as the predominant explanatory factors[3,5,8–18].

According to the climate-driven hypothesis of megafauna dynamics, a temporal dependency of population sizes on the glacial–interglacial cycle is expected. On the other hand, modern humans are expected to start influencing megafauna densities in recent times, mainly following the Last Interglacial period,

corresponding to their worldwide expansion out of Africa[19]. To distinguish between these two scenarios, previous studies have focused on inferring past species distributions and extinction chronologies based on fossil data[3,5,8–18]. However, while the fossil record provides valuable insight into species' histories, its fragmentary nature allows for only a limited temporal resolution of past population dynamics.

An alternative approach to fossil-based analyses is using genomic sequence data to reconstruct time-resolved trajectories of species population sizes[20,21]. Genomics-based methods commonly provide population size estimates for most of the Quaternary period (consisting of the Pleistocene period between 2.58 million and 11,700 years ago and the Holocene period between 11,700 years ago and present),

[1]Center for Ecological Dynamics in a Novel Biosphere (ECONOVO), Department of Biology, Aarhus University, DK-8000 Aarhus C, Denmark. [2]Center for Biodiversity Dynamics in a Changing World (BIOCHANGE), Department of Biology, Aarhus University, DK-8000 Aarhus C, Denmark. [3]School of Biology and Environmental Science, Faculty of Science, Queensland University of Technology, Brisbane, QLD, Australia. [4]Rewilding Europe, Toernooiveld 1, 6525 ED Nijmegen, The Netherlands. [5]Bioinformatics Research Centre, Aarhus University, DK-8000 Aarhus C, Denmark. ✉e-mail: jurajbergman@bio.au.dk

thereby covering multiple glaciation cycles, as well as recent periods of human expansion[22–32]. Thus, genomics-based trajectories of population sizes should provide a more comprehensive framework for modelling the impact of climatic shifts and humans on megafauna dynamics compared to fossil-based approaches. However, a global analysis of genomics-based megafauna histories and their driving factors is currently lacking.

We focus our study on the Late Pleistocene and Early Holocene population trajectories of extant megafauna to address the following hypotheses. On the one hand, the surviving species may have experienced similar dynamics as the species undergoing extinction, showing widespread population declines linked to *Homo sapiens* or climate. Alternatively, surviving megafauna communities may have exhibited compensatory dynamics[33], resulting in an increase in population size due to mechanisms such as competitive release. These scenarios have widely different ecological implications, whereby co-occurrence of population declines and extinctions would result in the exacerbation of ecosystem degradation, while compensatory dynamics would stabilise ecosystem functioning[34]. Thus, studying population dynamics of the surviving megafauna species during the Late Pleistocene and Early Holocene extinction period has major implications for our understanding of past and contemporary biosphere functioning[4,35].

We curated a genomic dataset comprising 139 high-quality reference genome assemblies and short-read sequence data of extant terrestrial megafauna and implemented a bioinformatic pipeline to infer their Quaternary population histories. We studied the population dynamics of megafauna as a function of species' ecology, geographical distribution, climate, and anthropogenic influence. We detect a global, severe decline in megafauna population sizes over the past 50,000 years and show that this observation is best explained by the influence of the worldwide expansion of *H. sapiens* rather than past climate dynamics. This lack of compensatory dynamics has had major impacts on ecosystem structure and functioning as reflected in a dramatic reduction of wild megafauna abundance, biomass and energy turnover.

## Results

### Severe decline of megafauna populations started during the late Quaternary

We implemented a bioinformatic pipeline to infer past dynamics of effective population sizes ($N_e$) in extant megafauna (Supplementary Data 1), with the time frame of estimates covering the Quaternary period (2.58 mya until present) for the majority of studied species. The pipeline consists of curating genome reference and short-read sequence data from existing databases, followed by read mapping and inference of genome-wide distributions of polymorphic sites for each species. The resulting segregating sites in a diploid genome are used as input to the pairwise sequentially Markovian coalescent (PSMC) method, based on a hidden Markov model parameterised with times and rates of coalescent events at each genomic locus[36], to infer average population sizes over discrete time windows in the past (see the "Methods" section). The resulting PSMC trajectories are transformed into effective population size changes over time (in years), using estimates of species generation times and mutation rates (Supplementary Note 1 and Supplementary Data 1). In total, we infer population dynamics from 139 megafauna species genomes (Fig. 1a) and observe a general decreasing trend towards the present time, as demonstrated by a positive correlation between effective population size and time before the present (Spearman's $\rho = 0.53$, $p < 0.001$).

To better characterise this decline, we fit a piecewise linear model to the estimated population size dynamics of species within different biogeographic realms, as well as a model of global population dynamics (Fig. 1a, Supplementary Note 2, Supplementary Fig. 2 and Supplementary Table 1). We identify two time periods with differing rates of population size change (henceforth, slope). The breakpoint

separating the two time periods is estimated to be within 32–76 kya across realms, with the global breakpoint within 48–52 kya. Realm-specific slopes for the period preceding the breakpoint are significantly negative for Africa and Eurasia, while positive and non-significant for Australasia and the Americas, respectively (Supplementary Table 1). In contrast, slopes become more severe and significantly negative across all realms for the time period after the breakpoint, indicating a global shift towards accelerating population declines closer to present time (Supplementary Table 1). Strikingly, the model of global population dynamics predicts a ~3.35% decrease in megafauna population size for the period between 50,000 ya and 1,000,000 ya, followed by an additional ~89.40% decrease over the last 50,000 years. Thus, during the last million years, more than 96% of the reduction in effective population size of extant megafauna occurred over the last 50,000 years.

We implemented a Bayesian framework (Supplementary Note 1) to estimate species-specific slopes over the entire time frame of their PSMC trajectories, while also taking into account the average adult mass of the species. The 95% highest posterior density interval (HPDI) of the slope is below zero for 91% (126/139) of the species, while 99% (138/139) of the species have a negative mean slope. The range of slope values varies between species and geographic regions (Supplementary Note 2), with the most severe slopes inferred for the Nilgiri tahr (*Nilgiritragus hylocrius*; 95% HPDI: [−0.715, −0.469]), Père David's deer (*Elaphurus davidianus*; 95% HPDI: [−0.671, −0.436]) and greater one-horned rhinoceros (*Rhinoceros unicornis*; 95% HPDI: [−0.632, −0.396]). Conversely, only the springbok (*Antidorcas marsupialis*) experienced an increasing, yet non-significant, population size trend (95% HPDI: [−0.127, 0.157]). Importantly, we recover a significantly negative relationship between species' mass and slope (Fig. 1b; 95% HPDI: [−0.152, −0.059]), indicating that larger species of extant megafauna also experienced stronger declines during the Quaternary, in line with the size-selection bias of recent megafauna extinctions[37].

We further investigate megafauna population dynamics by considering the ratio of lowest to highest population size (henceforth, decline severity), estimated for each species given its full PSMC trajectory. For 95% of species, we observe extremely strong decline severities ranging between 81.6% and 99.9% reduction in population size (Fig. 1c). To additionally characterise decline severity, we model it as a function of species' mass and time-point since a species experienced their lowest and highest population size (Supplementary Note 1 and Supplementary Fig. 6). Notably, the time-point since a species experienced its lowest population size is the main determinant of decline severity out of all predictors (Supplementary Fig. 7), despite the observation that 95% of species experienced their lowest sizes in a relatively short time window encompassing the last 40,000 years (Supplementary Fig. 6). This relationship suggests that population declines became more severe towards present time, reflecting unprecedented near-past population contractions in the majority of extant megafauna.

We next test the effect of phylogenetic relatedness between the studied megafauna species on the observed patterns in Fig. 1. To do this, we first repeat the analyses in Fig. 1a, c by subsetting our dataset to contain only one representative species per genus (67 species in total; Supplementary Note 3). In this way, we restrict our analysis to species that are unlikely to share evolutionary history (i.e. polymorphisms) throughout the time period of the inferred PSMC trajectories, thus minimising the phylogenetic signal in our dataset. When using this subset of more distantly related species, we again observe very similar patterns of megafauna decline (Supplementary Note 3 and Supplementary Fig. 8). We also test if the phylogeny of the studied species affected the relationship between species' adult mass and decline severity by conducting a phylogenetic regression analysis for a subset of species for which we were able to obtain a resolved phylogenetic tree (100 species in total; Supplementary Note 3). After controlling for

species' phylogeny, the re-estimated relationship between species' mass and decline slope remained largely unchanged (95% HPDI: [−0.167, −0.067]). We thus conclude that shared evolutionary history likely played only a minor role in shaping the relationship between body mass and decline.

Collectively, the strong trend towards population declines in extant megafauna, its co-occurrence with the Late Pleistocene and Early Holocene extinction period and mass-dependency of decline severity demonstrate a general lack of megafauna-mediated compensatory ecosystem dynamics during this period. In the next section, we therefore focus our study on the drivers of megafauna population dynamics as well as the consequences of the observed declines.

## Climate-based models are unable to predict population decline during the last 50,000 years

To better understand the recent population decline of megafauna, we focus our analysis on population trends during the last ~742,000 years (Fig. 2a) for which we have high-quality estimates of global temperature and precipitation dynamics[38,39]. Specifically, we are interested in whether climate conditions predict the recent severe declines in

megafauna population sizes. To test this hypothesis, we use population size estimates between 742 and 100 kya as the response variable to which we fit a model with climatic predictors. We then use this model to predict population sizes between 100 kya and the present, based on the corresponding climatic conditions of this time period. To represent climate, we focus on mean annual temperature and precipitation, as general climatic indicators with a clear link to the glacial–interglacial cycles characterising the time period studied. As predictors, we use the mean temperature and precipitation values of PSMC-inferred time windows, as well as the mean temperature and precipitation values of the preceding time window (i.e. climatic lag effect). Model fitting and prediction are conducted separately for each species (Supplementary Note 1).

In total, we fit 12 different climate-based models (Supplementary Table 10) and find that the model that assumes a linear relationship between both climatic predictors and the effective population size has the best predictive accuracy (Supplementary Fig. 10). Figure 2b and Supplementary Fig. 11a show the relationship between the predicted population sizes based on this model and the observed PSMC-estimated population sizes for four consecutive 25,000-year time

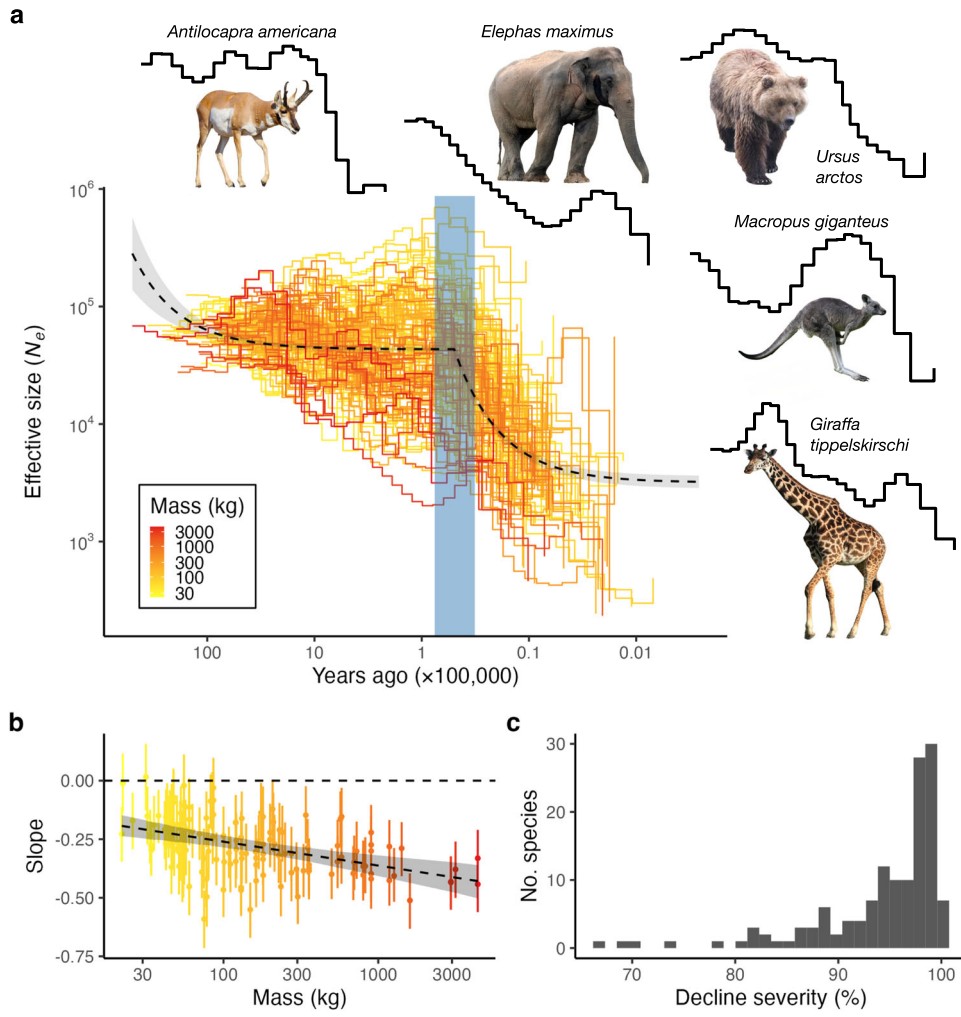

**Fig. 1 | Effective population size ($N_e$) dynamics of 139 extant megafauna species. a** Each step line represents changes in $N_e$ with respect to time for a single megafauna species, coloured by a gradient based on average adult mass. The dashed line represents the fit of the piecewise linear model, as determined by breakpoint analysis. The grey-shaded area represents the 95% confidence interval of the linear model prediction. The blue rectangle represents the timespan of realm-specific breakpoints (Supplementary Fig. 2). Both axes are log₁₀-transformed. Credit information for photographs of *Antilocapra americana, Elephas maximus, Ursus*

*arctos, Macropus giganteus* and *Giraffa tippelskirschi* are available in Supplementary Table 2. All photographs are under CC-BY copyright (https://creativecommons.org/licenses/by/4.0/) and adapted for the purpose of the figure. **b** Relationship between species' adult mass and the rate of population size change (slope). The *x*-axis is log₁₀-transformed. Points are median slope values with 95% HPDI ranges indicated by bars (each distribution is derived using *n* = 1000 posterior samples).
**c** Distribution of species' decline severity. Source Data for this figure are in Source Data 1–4.

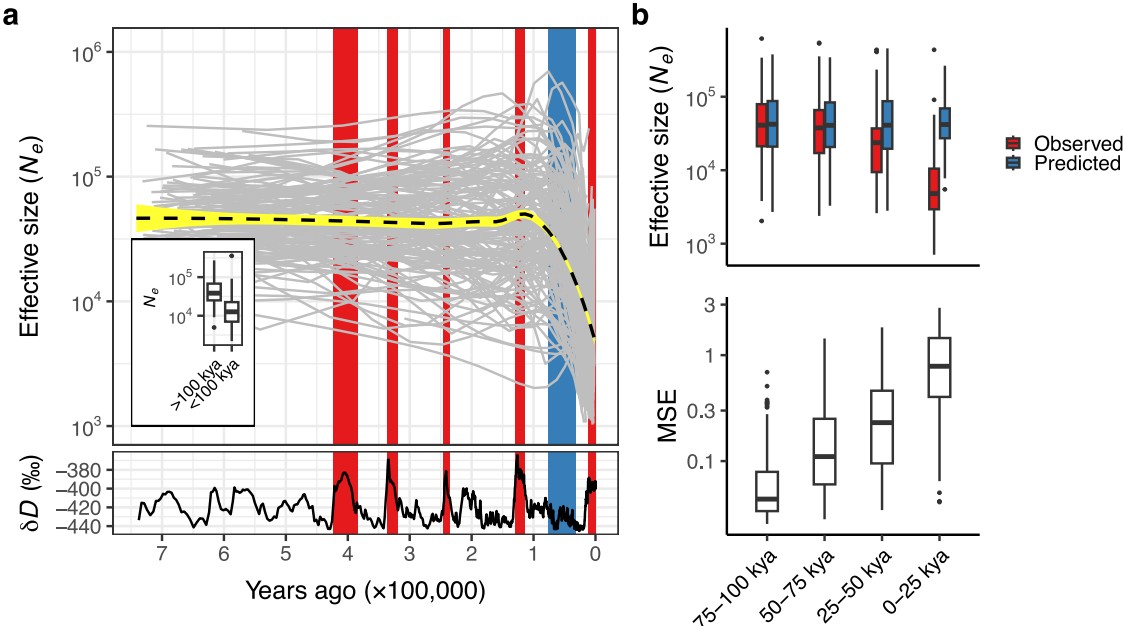

**Fig. 2 | Climate-based models of effective population size ($N_e$) trajectories.**
**a** Each grey line in the top panel represents an $N_e$ trajectory (log$_{10}$-transformed) with respect to time (in years) for a single megafauna species. The inset shows the distribution of average population sizes across species in the time periods prior to and after 100,000 years ago, respectively. Each box-plot contains $n = 139$ species-specific average population sizes. The median of the distributions is represented by the horizontal line within the boxes and box edges represent the interquartile range (IQR; 25th–75th percentile). The upper whisker extends from the upper box edge to the largest value no further than 1.5×IQR and the lower whisker extends from the lower box edge to the smallest value at most 1.5×IQR. Values outside whiskers are represented as individual points. The blue rectangle represents the timespan of realm-specific breakpoints estimated by the piecewise linear model (Supplementary Fig. 2). The black dashed line is the average population size trend across all species and the yellow ranges represent the 95% confidence interval, calculated using loess regression. The bottom panel shows temperature dynamics with warming periods highlighted in red. **b** Distributions of observed and predicted mean population sizes (top panel) and the mean squared difference between observed and predicted population sizes (MSE; bottom panel) across species for four time intervals during the last 100,000 years, estimated using the best-fitting climate-based model. Both y-axes are log$_{10}$-transformed. Each box-plot contains $n = 139$ species-specific values. The median of the distributions is represented by the horizontal line within the boxes and box edges represent the interquartile range (IQR; 25th–75th percentile). The upper whisker extends from the upper box edge to the largest value no further than 1.5×IQR and the lower whisker extends from the lower box edge to the smallest value at most 1.5×IQR. Values outside whiskers are represented as individual points. Source Data for this figure are in Source Data 5–9.

windows over the last 100,000 years. Notably, the difference between the observed and predicted values is larger for time windows that are closer to the present. The difference is non-significant for the oldest time window between 75 and 100 kya ($t$ [degrees of freedom = 139] = −1.054, $p = 0.294$, Cohen's $d = −0.089$, 95% Confidence Intervals = [−0.058, 0.018]), but gets progressively larger with proximity to the present (50–75 kya: $t$ [degrees of freedom = 139] = −2.807, $p = 0.006$, Cohen's $d = −0.238$, 95% Confidence Intervals = [−0.150, −0.026]; 25–50 kya: $t$ [degrees of freedom = 139] = −7.926, $p < 0.001$, Cohen's $d = −0.672$, 95% Confidence Intervals = [−0.370, −0.222]; 0–25 kya: $t$ [degrees of freedom = 139] = −26.063, $p < 0.001$, Cohen's $d = −2.21$, 95% Confidence Intervals = [−0.949, −0.816]; Fig. 2b and Supplementary Fig. 11a). This trend is also reflected in the increasing squared difference between the observed and predicted population sizes (i.e. mean squared error; MSE) during the last 100,000 years (Fig. 2b and Supplementary Fig. 11b). In conclusion, the detected time-dependency of model performance indicates the inability of climate dynamics to predict population shifts over the past 50,000 years.

**Models with human impact accurately capture recent population size dynamics**
To assess the explanatory power of climate and anthropogenic predictors on past megafauna dynamics, we consider 32 models (Supplementary Note 1 and Supplementary Table 10) with climate-only (12 models), human-only (4 models) or combined predictors (16 models made by combining predictors from 4 best-fitting climate-only models and all 4 human-only models). Predictors of human impact are based on estimated *Homo sapiens* arrival times to each biogeographic

realm[17], or, in the case of the Afrotropics, on *H. sapiens* establishment throughout the realm (Supplementary Table 3). We fit all models to the population size estimates of the last 742,000 years on a per-species basis and use leave-one-out cross-validation to compare model performance (Fig. 3a and Supplementary Fig. 12).

The model with the overall highest accuracy contains only human predictors and fits a logistic trend of megafauna population trajectories following human arrival. Given this model, the 95% HPDI for the rate of change in population size is negative for 68% (95/139) of megafauna species. Furthermore, the estimated median rate is negative for 93% (129/139) of species, indicating that megafauna species generally experienced a gradually accelerating decline in population size after human arrival. This is consistent with cumulative human impacts on megafauna populations post-arrival, as a consequence of the gradual establishment of human populations in a region. Human-only models with either a linear or exponential population size change had higher predictive accuracy than the majority of models with combined predictors and all climate-only models (Supplementary Fig. 12). Generally, the models based on annual temperature and precipitation predictors had the lowest log-scores and poorest predictive accuracy out of all tested models (Supplementary Fig. 12).

In Fig. 3b and Supplementary Fig. 13a, we show the distribution of MSE values across species for the best-fitting model in each class. Across the whole time span, the climate-only model had a significantly higher MSE compared to the human-only ($t$ [degrees of freedom = 1469] = 25.143, $p < 0.001$, Cohen's $d = 0.656$, 95% Confidence Intervals = [0.068, 0.079]) and combined models ($t$ [degrees of

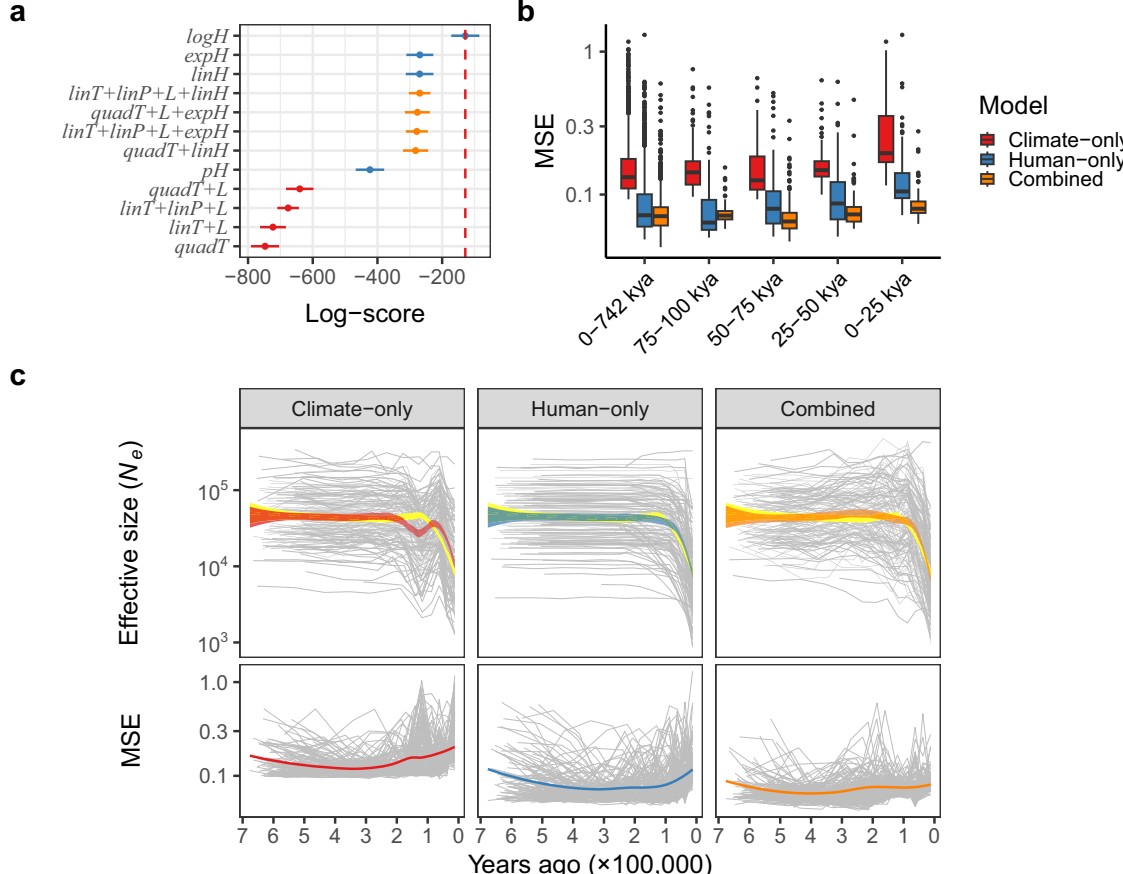

**Fig. 3 | Climate and human arrival-informed models of population size ($N_e$) trajectories. a** Log-scores of leave-one-out cross-validation for all climate and human-only models, and four best-fitting models with combined predictors. log*H*, exp*H* and lin*H* indicate models that assume a logistic, exponential and linear population trend following human arrival, respectively; *pH* indicates a model with the probability of human presence as the predictor; quad*T* and lin*T* (or *quadP* and *linP*) indicate models with a quadratic and linear temperature (or precipitation) effect on population trajectories, respectively; *L* indicates the inclusion of the temperature and precipitation lag predictor in the model (Supplementary Note 1). The red dashed line indicates the best-fitting model. Points are mean log-scores values with ±1 standard error indicated by bars (*n* = 1000). **b** Distributions of mean squared difference between observed and predicted population sizes (MSE) across species for the best-fitting model in each model class, for the whole time span (0–742 kya) and four time intervals during the last 100,000 years. The *y*-axis is log₁₀-transformed. For the whole time span

(0–742 kya), each box-plot contains *n* = 1470 species-specific $N_e$ values, while *n* = 139 species-specific $N_e$ values in each box-plot of the 25,000-year time intervals. The median of the distributions is represented by the horizontal line within the boxes and box edges represent the interquartile range (IQR; 25th–75th percentile). The upper whisker extends from the upper box edge to the largest value no further than 1.5×IQR and the lower whisker extends from the lower box edge to the smallest value at most 1.5×IQR. Values outside whiskers are represented as individual points. **c** The top panels show the observed and predicted population size trends of megafauna, given the best-fitting model in each model class. The yellow area is the mean observed population size trend, while each grey line represents the median predicted trend for a single species. The red, blue and orange areas are the mean predicted population size trends across species for the best-fitting climate-only, human-only and combined model, respectively. The bottom panels show the corresponding MSE values. Both *y*-axes are log₁₀-transformed. Source Data for this figure are in Source Data 10–12.

freedom = 1469] = 32.152, *p* < 0.001, Cohen's *d* = 0.839, 95% Confidence Intervals = [0.087, 0.093]). Despite a lower MSE value of the combined model compared to the human-only model (*t* [degrees of freedom = 1469] = 11.046, *p* < 0.001, Cohen's *d* = 0.288, 95% confidence intervals = [0.016, 0.023]), the lower log-score of the combined model (Fig. 3a) indicates that this model is prone to overfit the data, likely due to over-parameterisation of the model, and thus does not provide a generalisable explanation of the observed megafauna decline. We observe similar trends for each of the four discrete time windows during the last 100,000 years. Notably, the posterior predictive distributions across the last 742,000 years show that the largest discrepancies between the observed and predicted population sizes are present for time windows around the Last Interglacial period (116,000–129,000 years ago), especially for the climate-only model (Fig. 3c and Supplementary Fig. 13b). This is likely a consequence of the juxtaposition between the similarity in climatic conditions and dissimilarity in megafauna population sizes of the Last Interglacial and the

current warm period (Holocene; <11,700 years ago). The climate-only model therefore compensates between relatively high and low population sizes during the last two warming periods, respectively, by underestimating population sizes for the Last Interglacial, while overestimating them for the Holocene period. Additionally, the inconsistency in the ranking of best-fitting climate-based models when different time periods are considered (Fig. 3a and Supplementary Fig. 10) further points to the inadequacy of mean annual temperature and precipitation in explaining megafauna dynamics. In contrast, models that include human impact predictors showed much greater correspondence between mean observed and predicted population trends, lower variance of posterior predictive distributions and lower MSE across the whole time span (Fig. 3c and Supplementary Fig. 13b, c). We further test the influence of human expansion on megafauna dynamics by estimating the $N_e$ trajectories of human populations and calculating the correlation between human population sizes and the average megafauna population trend (Supplementary Note 4). We find

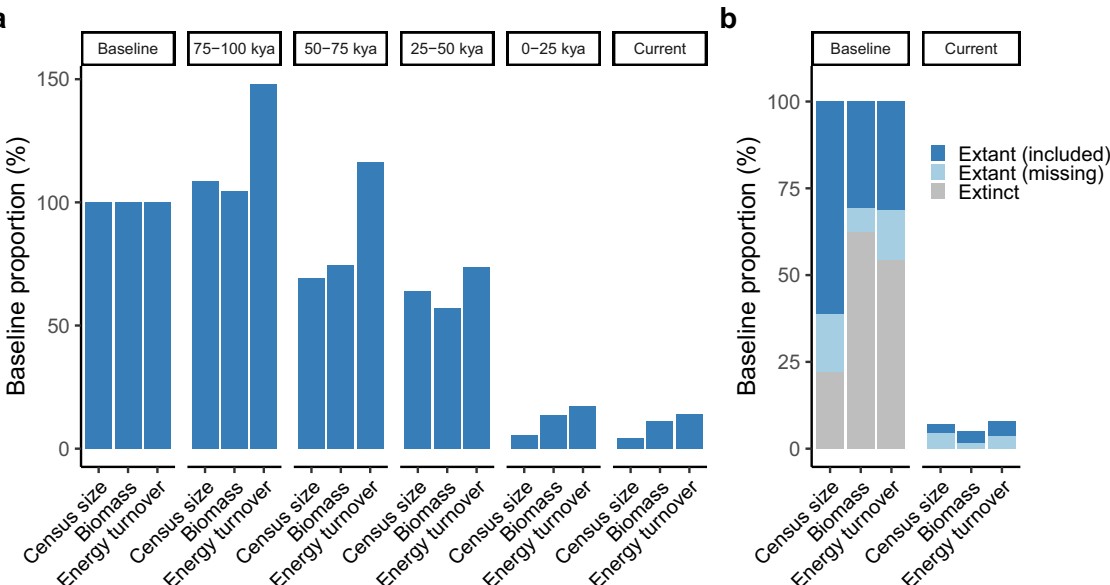

**Fig. 4 | Estimates of total megafauna individuals (Census size), biomass and energy turnover summed across species for different timepoints with respect to the baseline period. a** Parameter trends during the last 100,000 years with respect to the baseline period (100–742 kya) for 139 extant megafauna species that are included in our initial study dataset. **b** Contrast between the baseline and current period while taking into account all extant (both included and missing from our dataset) and extinct megafauna species (457 species for the baseline period and 260 species for the current period). Source Data for this figure are in Source Data 13, 14.

a strong negative correlation between human and megafauna PSMC trajectories for the time frame between 1.5 million and 150,000 years ago (Supplementary Note 4 and Supplementary Fig. 14). Over the last 150,000 years, human populations are characterised by extensive structuring and founder events[19], resulting in bottleneck dynamics in their PSMC trajectories between 150,000 and 50,000 years ago. However, between 50,000 years ago and the present time, human PSMC trajectories again exhibit population growth, in stark contrast to megafauna species which experience their lowest population sizes during this time frame (Supplementary Fig. 14).

## Consequences of megafauna decline

The negligible role of compensatory dynamics in offsetting the global megafauna extinctions, together with stark declines of the surviving megafauna populations during the last 50,000 years, is expected to have caused a drastic change in ecosystem composition and functioning[40]. To measure this effect in absolute terms, the effective population size ($N_e$) is inadequate, as it underestimates the total number of individuals in a population (i.e. census size; $N_c$), especially in wild populations[41,42]. We therefore use the ratio of the Holocene $N_e$ (as a proxy for the current effective size of a species) and the current IUCN census size estimates to relate $N_e$ values to estimates of $N_c$ (Supplementary Note 1). We then calculate the total megafauna census size, biomass and energy turnover (measured as the total daily metabolic rate) summed across species for different timepoints.

We focus the first set of analyses on parameter trends during the last 100,000 years for the 139 species in our dataset. Specifically, we assess the proportional change of megafauna parameter values for four time windows during the last 100,000 years, as well as for the current period (based on current IUCN estimates of census sizes; Supplementary Note 1), with respect to corresponding averages estimated for the baseline period (defined as the period between 100–742 kya). For the period between 75 and 100 kya, we observe a slight increase in total megafauna census size and biomass (~9% and ~4%, respectively) and a marked increase of ~48% in energy turnover, compared to the baseline period. This difference is driven by a slightly higher total number of megafauna individuals and a shift in relative

species abundance characterised by a lower contribution of larger species to the total megafauna census, biomass and especially energy turnover for the period between 75 and 100 kya. Importantly, we observe a continuous decline in all parameters across the last 100,000 years, with total census size and biomass decreasing below the baseline between 50 and 75 kya, while energy turnover decreased below the baseline between 25 and 50 kya (Fig. 4a). At the current time, the percent change from the baseline reached a reduction of -96%, -89% and -86% for total census size, biomass and energy turnover, respectively. This reduction equates to a total loss of ~660 million megafauna individuals, ~0.008 gigatonnes of carbon (Gt C) of biomass and a ~2.4 pJ/day of energy turnover compared to the baseline period. Together, these results indicate that the current period is climatically suitable for accommodating a much greater number of wild large animals with much greater ecological effects than are present in contemporary ecosystems.

To obtain a complete overview of the impact of megafauna decline, we use the estimated relationship between species' census size and mass in our dataset to infer parameters of ecological importance for extinct species, as well as extant megafauna species that are missing from our dataset (Fig. 4b and Supplementary Note 1). We estimate that the global megafauna census was ~1.1 billion individuals across a total of 457 megafauna species (139 extant species comprising our initial study dataset, 121 extant species missing from this dataset and 197 extinct species; Supplementary Note 1) during the baseline period, which declined by ~93%, i.e., to ~80 million individuals during the current period. Similarly, total baseline biomass (~0.03 Gt C) and energy turnover (~8.9 pJ/day), declined by 95% (currently, ~0.001 Gt C) and 92% (currently, ~0.70 pJ/day), respectively. Furthermore, the proportion of total megafauna census, biomass and energy turnover contributed by extinct species during the baseline period is estimated to be ~22%, ~62%, and ~54%, respectively, suggesting that these species constituted the primary functional megafaunal component of past ecosystems. Estimating past $N_c$ from $N_e$ involves substantial uncertainties. We thus also explore alternative estimation approaches; however, they all provided broadly similar estimates (Supplementary Note 1 and Supplementary Note 5). Similar results are also obtained

when directly comparing patterns of megafauna loss of the current period with respect to the baseline estimated for the climatically similar Last Interglacial period[43,44] (Supplementary Note 6).

Another important consequence of population decline is an increase in the current extinction risk of a species. We study the extinction risk of megafauna by considering the relationship between their baseline effective population size, calculated as the average $N_e$ for the period between 100–742 kya, and their census size estimated by IUCN (Spearman's $\rho = 0.412$, $p < 0.001$, Supplementary Fig. 23). Generally, higher census than effective sizes are observed across various animal groups, with the median $N_e/N_c$ ratio estimated to be ~0.10–0.14[41,42]. On the other hand, a lower $N_c$ compared to $N_e$ is expected in species that have undergone recent population bottlenecks and have a high current extinction risk[45,46]. Strikingly, the median ratio of baseline $N_e$ to the current census size for megafauna in our dataset is estimated to be 0.98, with 49% (49/99) of species having a lower census size compared to their baseline effective size. Additionally, species with a higher $N_e/N_c$ ratio tend to have higher adult mass (Spearman's $\rho = 0.199$, $p = 0.048$; Supplementary Fig. 24), again signifying stronger population declines experienced by larger megafauna. The failure to recover census sizes in these species is likely to cause further genetic degradation of their populations and lead to an increase in their extinction risk and elevated risk of disruptions of the ecosystems they inhabit.

## Discussion

Our results show that megafauna communities have experienced severe declines over the last 50,000 years, not just through extinction[1-3], but also through severe reductions in the population sizes of surviving species. Analogous to the strong size-selectivity of the extinctions[37], the population declines were most severe for larger species (Fig. 1b). We also show that this downsizing was unique relative to earlier periods of the Quaternary and that human presence was likely the driving factor, as opposed to climatic changes. Given the compounding effect of megafauna extinctions, population size reductions of extant megafauna and lack of megafauna-mediated compensatory dynamics, it is clear that the Late Pleistocene and Early Holocene periods witnessed a major restructuring of ecosystems at a global scale, leaving current ecosystems in severely megafauna-poor states relative to the Quaternary norm.

Inference of population trajectories using PSMC methodology emphasises the use of population genomics as a tool to study determinants of long-term species dynamics and potentially inform conservation and restoration targets[47]. It should also be noted that PSMC trajectories can be influenced by past population subdivisions and migration patterns[48], thus affecting the interpretation of population size changes inferred by PSMC. However, it is unlikely that the population structure of studied megafauna species (which differ with respect to their life history traits, locomotive ability, habitat preference and geographic distributions) would have been similar enough to manifest as a global decline with an onset during a very narrow time window. As this time window largely corresponds to human arrival times to each realm (Supplementary Fig. 2), the global expansion of *H. sapiens* remains the most parsimonious explanation for the decline of extant megafauna.

The Late Pleistocene and Early Holocene extinctions resulted in multiple co-extinctions and reduction of diversity due to the loss of important ecological roles performed by these species[40]. Although such events might have provided opportunities for population expansion in surviving species through compensatory dynamics, the observed decline of extant megafauna during this time indicates that such a scenario was never realised. Importantly, this does not exclude the possibility of partial compensation through smaller or moderately sized species, as indicated by the dependence of decline severity on mass (Fig. 1b), as well as observed population increases in some species

of birds[49], bats[50] and insects[51,52]. Furthermore, our results suggest that, beyond local and regional extirpations, extant megafauna suffered significant declines in population density. Breakpoint analysis (Supplementary Fig. 2 and Supplementary Table 1) showed that mild megafauna declines were present in Africa and Eurasia prior to *H. sapiens* expansion. These results could be a consequence of archaic, pre-sapiens *Homo* presence in Afro-Eurasia[19,53,54] or an effect of the repeated and somewhat intensifying glaciation cycles[55]. However, contradicting such a climatic explanation, Australasia and the Americas had stable and even slightly increasing megafauna populations during this time frame. We further observed that the shift in megafauna dynamics in the Americas preceded known human arrival dates (Supplementary Fig. 2), which could reflect either increasing cooling towards the glacial maximum (most of these species are warmth-adapted species) combined with later anthropogenic suppression of populations, or earlier human arrival dates[56] (but see ref. [57]).

Differences in megafauna decline patterns between land masses could be underlied by region-specific intensity of global climatic events, as suggested for the Younger Dryas cooling episode (12,900–11,300 years ago)[58,59]. To test the effect of such climatic episodes on megafauna dynamics would require population size estimates at finer temporal scales than the ones provided by the PSMC method. Additionally, a more nuanced consideration of climatic parameters other than the mean annual temperature and precipitation, as is the focus here, might yield higher explanatory power of climate-based models of population dynamics, which would be a suitable analysis for future studies with higher temporal resolution of population size changes. However, given the relatively large (25,000-year) windows for which we conduct our analyses, we found a general inability of climate-only models to predict population declines (Supplementary Figs. 11 and 13), as well as the increased performance of models that include predictors based on human arrival (Supplementary Fig. 12), indicating that humans played a dominant role in the global contraction of megafauna populations.

The inference of long-term population dynamics allowed us to provide estimates of past census sizes, biomass and energy turnover of extant and extinct megafauna (Fig. 4). Strikingly however, the current total biomass of the entire current wild terrestrial mammal community, estimated by Bar-On et al.[60] to be ~0.003 Gt C, is 10% of our estimate for the total megafauna biomass during the period between 100–742 kya (~0.03 Gt C), implying that mammals—both only considering still extant species or combined with extinct species—played a much greater role in past ecosystems compared to present time.

Importantly, given that our estimates of past $N_c$ values are based on current $N_c$ of populations that experienced severe recent contractions, we likely underestimated past megafauna $N_c$. To gauge further insight into past megafauna dynamics and the potential extent of this underestimation, we can refer to historical estimates of megafauna population sizes during time periods that precede intense global industrialisation. Two of the most prominent examples of pre-industrial megafauna populations come from 19th century direct observations of the North American bison population[61], and estimates based on 19th and early 20th century ivory trade for the African elephant[62]. These studies report population sizes of approximately 30 and 27 million individuals for the bison and elephant, respectively. Considering these estimates, we conducted an additional analysis with a modified $N_e/N_c$ ratio that we then used to re-estimate megafauna census sizes through time (Supplementary Fig. 15 and Supplementary Note 1). This analysis resulted in an estimate of ~95 billion megafauna individuals, ~3 Gt C of biomass and ~850 pJ/day of energy turnover during the baseline period. These estimates could have been greater still, as megafauna population sizes were likely already suppressed at the time of the 19th and early 20th century estimates. On the other hand, between-species competition and restrictive environmental conditions could have acted as potential limiting factors of achievable

megafauna densities, and thus negatively affected megafauna population sizes during the baseline period. Similar estimates were obtained using alternative approaches of $N_c$ estimation (Supplementary Note 5).

At present time, megafauna populations have decreased to low abundances[63] and a large fraction of surviving megafauna are threatened with extinction[64–66], casting further uncertainty on the future of these species and their ecosystem functions, such as herbivory- and disturbance-linked promotion of heterogeneity in vegetation and soil[67,68], plant and nutrient dispersal[69,70], and trophic interactions among mammals and co-dependent species[40,71]. Importantly, our results indicate that the current epoch could support substantially greater megafauna biomass than typically assumed, given the similarity of the current warm period to the Last Interglacial[43,44] (Supplementary Note 6). The fulfilment of this potential through trophic rewilding[72] would require planning at a global scale and strong upscaling of current conservation and restoration efforts[6,7].

## Methods

### Data curation

Reference genome assembly and short-read data accessions for the 139 species used in this study were downloaded from public databases, which included the National Center for Biotechnology Information (NCBI; https://www.ncbi.nlm.nih.gov/), the European Nucleotide Archive (ENA; https://www.ebi.ac.uk/), the DNA Zoo database (https://www.dnazoo.org/), the GigaDB database (http://gigadb.org/), the National Genomics Data Center (NGDC; https://ngdc.cncb.ac.cn/), the Broad Institute (http://ftp.broadinstitute.org/) and http://www.caribougenome.ca/. Per-species accessions of fastq files, reference sequences used for mapping, and their source databases, are listed in Supplementary Data 1. For short-read mapping, we chose short-read data from one representative biosample per species, corresponding to the individual used for reference assembly or, when these data were unavailable, we chose a biosample of the corresponding species with a large dataset of short reads. We also searched the databases for short-read data of individuals that are representative of different populations (or subspecies) within the species complex and included them in our analysis. When data of multiple individuals were available from a single population, we chose the individual with the largest amount of short-read data as a representative, such that we maximise genomic coverage of the mapped bam file and thus ensure high accuracy of the population size estimates. Temperature data were taken from Augustin et al.[38] and consisted of estimates for the last 742,419 years (abbreviated to 742,000 years in main text). Precipitation data was taken from the pastclim database[39] and processed by calculating the global mean of the annual precipitation parameter (variable bio12 in the pastclim database) across 1000-year windows spanning the last 800,000 years[73].

Each species was assigned to one biogeographic realm and biome, as defined in Olson et al.[74]. To do this, we considered the overlap of the species' geographic range, estimated using the PHYLACINE database[75], with each of these geographic classifications. If a species' range overlapped multiple realms (or biomes), the assignment was conducted by choosing the realm (or biome) with the largest overlap. Additionally, megafauna subspecies that were not listed in the PHYLACINE database were assigned to the realm (or biome) of the closest related species with available data. An analogous procedure was implemented when assigning species to human biogeography regions, which were taken from Sandom et al.[3]. Geographic classification of all species is presented in Supplementary Data 2 and additional analyses of these classifications are presented in Supplementary Note 2.

Selection of species for our dataset was conducted such that multiple representative species of extant megafauna were included for each biogeographical realm. The smallest included megafauna species was the western grey kangaroo (*Macropus fuliginosus*; 22 kg in adult weight), as an extant megafauna representative of the Australasian

realm. The median adult mass of species was 120 kg across the full dataset. Species' adult masses were taken from PHYLACINE and metabolic rates (energy turnover) were taken from Pedersen et al.[76]. We expressed total biomass values in units of gigatonnes of carbon (Gt C), by assuming that adult mass consists of 15% carbon, as in Bar-On et al.[60], while total energy turnover was expressed in units of petajoules per day (pJ/day).

*Homo sapiens* arrival ranges were classified with respect to biogeographic realm and taken from Andermann et al.[17], except for the Afrotropic realm, where the arrival range corresponded to the timeframe of *H. sapiens* establishment throughout the Afrotropic realm (Supplementary Table 3).

### Mapping of short-read data

To process fastq files (accessions in Supplementary Data 1) into mapped bam files needed as input for the pairwise sequentially Markovian coalescent (PSMC) programme (https://github.com/lh3/psmc)[36], we followed the best practice workflow for data pre-processing of the Genome Analysis Toolkit (GATK)[77]. We first processed the fastq files using picard tools (https://broadinstitute.github.io/picard/) to generate unmapped bam files (using the FastqToSam module), which are the required input for the MarkIlluminaAdapters programme. This programme allowed us to mark and remove Illumina adapter sequences from reads and thus avoid adapter-related biases during the read-mapping process. The programme bwa mem v0.7.17[78] was then used to map the reads to reference sequences using default settings. All reads were mapped to previously published reference sequences whose accessions and source databases are listed in Supplementary Data 1. When the exact reference was unavailable or of low quality for a particular species, we used the reference of a closely related species of sufficient quality, i.e. with a scaffold N50 value (the minimum size of scaffolds that contain more than 50% of the assembled reference) larger than 20 kb. The median N50 value of references used for mapping was 66.6 Mb across species, with references of only 13 species having an N50 lower than 1 Mb (Supplementary Data 1). Only reference scaffolds that were more than 1 kb in length were used for mapping of reads. Secondary alignments and duplicates were removed using Picard tools MergeBamAlignment and MarkDuplicates modules. The resulting mapped bam files were sorted by coordinates using the Picard SortSam module, followed by indexing with the samtools index module. In species for which short read data were spread across multiple accessions (Supplementary Data 1), we merged the resulting bam files into the final bam file using the Picard tools MergeSamFiles module. Additionally, coverage of genomic positions was calculated using the samtools depth programme[79] (the median genomic coverage across species was 45×; Supplementary Data 1).

### Demography inference

For demography inference, we used the pairwise sequentially Markovian coalescent (PSMC) implementation (https://github.com/lh3/psmc)[78]. To account for potential inference biases introduced by genomic regions with low mapping probability, we created a mappability filter for each reference genome using the snpable programme (http://lh3lh3.users.sourceforge.net/snpable.shtml), with the 90% stringency criterion as in Palkopoulou et al.[24]. We used the bcftools[79] mpileup programme to produce pileup files from the mapped bam files using only sites that passed the stringency criterion of the mappability filter, had minimum mapping quality (MAPQ) of 20 and PHRED base quality of 20. The pileup files were then used as input into the bcftools call module (in "-c" mode for consensus calling and using the "-V indel" option to exclude insertion and deletion variants). The produced bcftools output files were used as input into the vcf2fq module of vcfutils[79] to produce consensus sequences for each species. We set the minimum ("-d") and maximum coverage ("-D") options of the vcf2fq module such that we included only those sites whose

coverage was not below 1/3 or above twice the average genomic coverage of the species (Supplementary Data 1), in order to further filter out genomic sites covered by reads with potential sequencing or mapping biases. The resulting fq files were used to create PSMC input files using the fq2psmcfa tool (with option "-q20")[36]. As longer sequence length increases the accuracy of PSMC inference[80], we used only scaffolds of at least 100 kb in length for demography inference. Additionally, we used all genomic sites that passed the aforementioned criteria for inference, thus assuming that the majority of the remaining segregating sites evolve under neutrality (or close to), as might be expected under the assumption that the majority of positively or negatively selected sites are fixed (non-segregating) in the population. We ran the PSMC programme with three different settings for the "-p" parameter ("$4 + 25 \times 2 + 4 + 6$", "$6 \times 1 + 24 \times 2 + 4 + 6$" and "$10 \times 1 + 15 \times 2$") and selected a single PSMC output per species (Supplementary Data 1 and Supplementary Data 3) that maximised the number of recombination events used to estimate effective population sizes ($N_e$) in each time interval[36].

Conversion of the PSMC output into effective population sizes and time (measured in years) was done following https://github.com/lh3/psmc[78]. The per-generation mutation rate for each species was obtained from literature or predicted using a regression model (Supplementary Fig. 1) based on known mutation rates and generation times of extant mammals[81], as described in Supplementary Note 1. Publication sources for mutation rates and generation times of each species are listed in Supplementary Data 1.

### Statistical modelling
Breakpoint analysis used to determine the time range for which population size change became more severe was conducted using the "segmented" library[82] implemented for the R programming language. Population size estimates were $\log_{10}$-transformed prior to breakpoint analysis.

Statistical modelling of parameters that impacted megafauna population trajectories and estimation procedures of total megafauna census sizes, biomass and energy turnover is described in detail in Supplementary Notes 1 and 5. A list of response and explanatory variables used in models, along with their description, is presented in Supplementary Table 13. All models were fitted using a Bayesian framework implemented in the probabilistic programming package pyMC3 of the Python programming language[83]. All models were run using four Markov chains, each with 2000 tuning iterations followed by the same number of sampling iterations to infer posterior parameter distributions. Posterior sample distributions were constructed using 1000 randomly sampled iterations from the posterior parameter distributions. Leave-one-out cross-validation of the fitted models was conducted using the Python-implemented ArviZ package[84].

### Reporting summary
Further information on research design is available in the Nature Portfolio Reporting Summary linked to this article.

## Data availability
The data and scripts used in this study are deposited in GitHub (https://github.com/jbergman/megaFaunaHistories) and Zenodo repositories (https://doi.org/10.1101/2022.08.13.503826). The raw data used in this study are available from public repositories—data accession codes are listed in Supplementary Data 1. Source data are provided with this paper.

## Code availability
Code used in this study is deposited in GitHub (https://github.com/jbergman/megaFaunaHistories) and Zenodo repositories (https://doi.org/10.1101/2022.08.13.503826). Additional code used in the analysis is available on request.

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

## Acknowledgements

This work was supported by J.-C.S.' VILLUM Investigator project "Biodiversity Dynamics in a Changing World", funded by the VILLUM FONDEN (grant 16549) and Center for Ecological Dynamics in a Novel Biosphere (ECONOVO), funded by the Danish National Research Foundation (grant DNRF173 to J.-C.S.). E.A.P. was supported by the project TERRANOVA, the European Landscape Learning Initiative, which has received funding from the European Union's Horizon 2020 research and innovation programme under the Marie Sklodowska-Curie grant agreement no. 813904. The output of the manuscript reflects the views only of the authors, and the European Union cannot be held responsible for any use that may be made of the information contained therein. We thank Genome DK (https://genome.au.dk/) for bioinformatics support.

## Author contributions

J.B. carried out data collection, primary analysis and wrote the initial draft of the manuscript. J.-C.S. and M.H.S. provided the initial conception of the study. J.B., R.Ø.P, E.J.L., R.T.L., S.M., E.A.P, M.H.S. and J.-C.S. contributed to model development, interpretation of results, and subsequent drafts and revisions of the manuscript.

## Competing interests

The authors declare no competing interests.
