## [Peer Review File · Nature Communications]

Worldwide Late Pleistocene and Early Holocene population declines in extant megafauna are associated with Homo sapiens expansion rather than climate changeReviewers' Comments:

Reviewer #1:

Remarks to the Author:

In this manuscript Bergman et al present an interesting and novel insight into the population size dynamics of 139 mega-fauna species throughout the last 1 million years. The trends of effective population sizes are estimated from genomic data from individuals of these species at the present, utilizing population genetic models and full genome sequences. They find a correlation between the timing of the detected population size decreases and the worldwide expansion of *Homo sapiens*, while the climatic fluctuations during this time window do not appear to have left a signature in the population size trajectories of these species.

The article is well written and adds another puzzle-piece to our understanding of the dynamics of recent declines of megafauna species and the potential role our own species may have played in that decline. The data seems to support the conclusions and the methodological approach is quite creative by combining population genetic methods with macroecological analyses. However, some more methodological detail and discussion is needed to really help to convince readers of the validity of the results and the appropriateness of the applied methods (see comments below).

To generate some more perspective on the method of population size modeling from current DNA patterns, it would be illustrative to also show in comparison to these species in Figure 1 how human effective population size changes in the same time window, using the same method. I'm curious if for *Homo sapiens* the estimated effective population size through time would match the trends that we know of in terms of actual population size (census size). If they don't match, I still (or even more so) think it is important to display the human N_e curve in this plot, so readers can appreciate that there can be differences in the trends between actual population size and effective population size estimates from genetic loci.

A very essential component of this paper is how the genetic data were processed and analyzed to get to the population-size-through-time estimates. The methods description on this essential and complex element of this study is currently too short (lines 380-411) and lacks important information, making it difficult to impossible to evaluate how well the results are supported by the data. I could not find any more detailed description of these methods in the supplementary material (apologies in case I missed something). See e.g. <https://doi.org/10.1093/molbev/msu058> for an example of a similar methodology described in appropriate detail.

Some specific points that definitely need more information are the following:

- line 383: what is the purpose of creating "unmapped bam files with marked adapter sequences"? Was this to remove any remaining Illumina adapter contamination? The purpose is not clear, particularly since in the next step the reads are mapped to references, creating mapped bam files.

- line 384: what were the reference sequences used for read mapping? This is very relevant information! The next sentence seems to imply that they were mapped against contigs of a certain size, but it is not clear where those contigs are coming from. Did they come from the same sample via *de novo* assembly or from some reference taxon that was used? If it is the former, that step needs to be described (how were the contigs generated). After reading through the methods again I found some of the relevant information in the lines 351-355. This text should be moved to go together with the rest of the sequence data methods descriptions, otherwise it's hard to connect the dots.

- can you provide any information on the nature of the genetic loci that were used here? were they always the same for all species, are they just randomly selected from the genomic data and are there any assumptions that can be made about them being neutral markers or under selection? These dynamics matter for effective population size estimation and should we touched upon (can be in the

supplementary material).

It should be emphasized more in the main manuscript how the population size through time curves were dated (via estimates of mutation rates for individual species and the detected number of mutations/heterogenous sites in the genetic data). This is key to understanding this study and should be made clear. Supplementary text 1 covers this part in some more detail, but this information deserves a place in the main text of the manuscript. Since the authors are using genome-wide genetic data, I wonder how a single mutation rate can account for the vast variation of mutation and substitution rates present across the genome. There are vast differences in these rates across different loci. It is obviously not feasible/possible to find or produce rate estimates for every genetic locus represented in the data, but this limitation should be highlighted and discussed. In the current version I could not find any such discussion in the main text or the supplementary text.

I wonder about the use of the terminology "mutation rates". Wouldn't substitution rates be the more appropriate term here, since the authors are looking at the number of fixed nucleotide polymorphisms (substitutions)?

Lines 69-71: The authors refer to the identified "lack of compensatory dynamics". Given that only species above 22kg were considered as megafauna species in this study, would it be conceivable that medium-sized or smaller species that occupy overlapping ecological niches with the disappearing megafauna increased in population sizes and may have displayed some kind of compensatory dynamics? Just because these can't be detected in the megafauna species, it does not mean that such ecological compensation did not happen at all. This should at least be acknowledged or discussed.

Line 18: "similar declines": I suggest to be more precise in the wording here, because "similar" would mean going extinct.

Fig 1: The giraffe image has a copyright-watermark on it. Check carefully the copyright license for all these pictures and cite the source where appropriate. I suggest to turn the x-axis in panel A into real numbers (1,000, 10,000, 100,000), to increase readability.

Lines 75-81: Reading this paragraph makes me immediately curious how the dating of the population dynamics was done. I think that is relevant information for the understanding of these results and should be mentioned here (can be a single sentence, briefly mentioning the dating method).

Line 304: Remove the ", indicating"

Reviewer #2:

Remarks to the Author:

In this report, Bergman et al. concisely and clearly report estimates of historical population sizes for a diverse group of megafauna, demonstrating a link between this decline and human dispersal. The piece is superbly written and the science rigorously executed.

While the article is excellent, I would like the authors to reflect on the possible effects of population structure, multiple testing, and phylogeny on their inferences. It would be interesting to discuss the extent to which increasing population structure through time - associated with population declines - might have impacted the PSMC results.

Similarly, the authors might want to comment or directly address the impact of phylogeny versus body size on their results. This could be done via phylogenetic regression. Concretely, if all large animals are also very closely related, it is difficult to point to body mass over other inherited traits that might increase their chances of population decline. Similarly, body mass is associated with a broad range of

life history traits, so it might be worthwhile to at least speculate on what might be the root cause of the susceptibility of large species to population decline in the hands of early human societies. For instance, small clutch sizes might be the ultimate cause of a slow replenishing of populations.

The authors might have also addressed already the repeated tests using of temperature data across species. It would be worth making sure that such a correction is implemented and/or described in the text.

Reviewer #3:

Remarks to the Author:

Noteworthy results:

The most noteworthy results of the analyses presented are a that there is a significant relationship between the timing of human arrivals across continents and the extinction of megafauna, but no significant relationship with climate. Additionally this work presents estimates for megafauna census sizes, biomass and energy turnover between the Last Interglacial and the current period. Overall these estimates make for a compelling argument of the severe ecological impacts of end Pleistocene megafauna extinctions.

Significance to the field and related fields:

I believe that this work has the potential to be significant to fields that take interest in late Pleistocene extinctions as well as modern conservation. It does not present an overall result that is new, the consequences of the loss of megafauna is widely discussed in this field and the debate over human vs. environmental causes is many decades old. However, this paper offers quantification of the consequences of megafauna extinction that are striking and the overall approach, with some caveats, is sound.

Support for the conclusions and claims

The work supports the conclusions and claims, however there are concerns with how the authors have used some previously published data from other investigators for their purposes.

Flaws in the data analysis, interpretation and conclusions that require revision:

My most significant critique of this study is the use of previously published datasets beyond what is appropriate and the resulting overinterpretation of these data. While I am inclined to agree with the authors that climate is probably not playing the primary role in continental/global scale patterns of extinction, this paper severely overstates what they have uncovered about climate's role in driving effective population sizes, or lack thereof. There is an abundance of literature documenting AET, min/max temperature, or CV of temp (among others) as better predictors of diversity and distribution of species than even regional MAT. These aspects of climate don't necessarily tightly covary with MAT, so MAT is not a catch all or stand in for "climate." What the authors have demonstrated is that global MAT is not a driver, and they can confidently state that. However extrapolating their findings to all aspects of climate, which they appear to be doing, is a reach. Furthermore, there needs to be some justification provided for the extrapolation of MAT taken from a singular Antarctic ice core to the entirety of the globe. I find this concerning because we know that there can be large differences in the records documented between ice cores and regional proxy records that are more appropriate for the "realms" being investigated in this study. For examples of the variation I am referring to see Fiedel 2011 in Quaternary International and Fastovich et al. 2020 in Geophysical Research Letters. This problem is especially pertinent to lines 325-327 in the discussion.

Soundness of methodology:

The approach is logically sound and is clever, however I stand by my critique of their use of previously published data.

Reproducibility of the analyses:

This paper uses an approach that is purely mathematical and there appears to be enough detail provided in the methods for the work to be reproduced. It would be of value for future work to use their approach with alternative datasets that lack the issues I have laid out.

Other specific comments:

Line 17: This dichotomy is increasingly seen as a strawman argument, there are a growing number of paleoecologists who hold a more nuanced perspective that both human and climate have varying degrees of impacts on extinct megafauna at different scales

Figure 1: Some of the animal icons in figure 1 have clear copyright watermarks on them, please ensure that appropriate rights are obtained for images and figures.

Line 139: The particulars of how climate plays out on landscapes, especially in the interior of continents, can vary considerably from what is documented in ice core data. In other words, it may not reflect the climate conditions that are experienced by organisms.

Lines 158 - 159: Global temperature records are widely available, making them quite convenient, but we know that ecosystems respond to different aspects of the climate beyond mean annual temperature. In many cases precipitation is as influential, or more so, than temperature. For this reason, it's not all that surprising that a significant pattern isn't observed with MAT.

Lines 236-239: Data that has been log-normalized has variance that is compressed relative to the raw values and yet the strength of this correlation is in the mid-range of what might be considered as only "moderate." This should be acknowledged, and I would like to see the authors clearly explain how this impacts the calculated values of total megafauna census size, biomass, and energy turnover.

Line 304: ", indicating." this looks like a typographical error.

Lines 344-346: How similar IS the current interglacial to the last interglacial? Such a claim needs references.

Line 353: How was it determined if a sample was representative? Ne can vary greatly regionally and across populations, how is this addressed?

Reviewer #1 (Remarks to the Author):

In this manuscript Bergman et al present an interesting and novel insight into the population size dynamics of 139 mega-fauna species throughout the last 1 million years. The trends of effective population sizes are estimated from genomic data from individuals of these species at the present, utilizing population genetic models and full genome sequences. They find a correlation between the timing of the detected population size decreases and the worldwide expansion of *Homo sapiens*, while the climatic fluctuations during this time window do not appear to have left a signature in the population size trajectories of these species.

The article is well written and adds another puzzle-piece to our understanding of the dynamics of recent declines of megafauna species and the potential role our own species may have played in that decline. The data seems to support the conclusions and the methodological approach is quite creative by combining population genetic methods with macroecological analyses. However, some more methodological detail and discussion is needed to really help to convince readers of the validity of the results and the appropriateness of the applied methods (see comments below).

To generate some more perspective on the method of population size modeling from current DNA patterns, it would be illustrative to also show in comparison to these species in Figure 1 how human effective population size changes in the same time window, using the same method. I'm curious if for *Homo sapiens* the estimated effective population size through time would match the trends that we know of in terms of actual population size (census size). If they don't match, I still (or even more so) think it is important to display the human N_e curve in this plot, so readers can appreciate that there can be differences in the trends between actual population size and effective population size estimates from genetic loci.

Answer: We agree and have now conducted an additional analysis of the *Homo sapiens* PSMC population trajectory and its relation to megafauna population trajectories (Supplementary text A). For this, we followed the approach described in Chen et al. (2019; 10.1126/science.aav6202), where authors infer a PSMC trajectory using a hybrid African-Asian sequence simulated from the best-fitting model of human demography estimated by Schaffner et al. (2005; 10.1101/gr.3709305), and correlated this trajectory with PSMC trends of ruminants species. The combined human sequence is meant to mimic the global expansion of *Homo sapiens*, given that it contains a global representation of genetic variation accumulated in different human populations - in this case, the African and Asian sequences were chosen as they are the most geographically and genetically diverged populations considered by Schaffner et al. (2005). We calculated the correlation of the human PSMC trajectory against the median megafauna population trajectory and displayed the results in Supplementary text A: Fig. 1. The correlation coefficient between the human and megafauna population trajectories was estimated to be significantly negative (Spearman's $\rho = -0.82$, $p < 0.001$), in concordance with a negative human influence on megafauna population dynamics, as also observed in Chen et al. (2019; their Fig. 2A; please note that the time axis of their plots is reversed compared to the plots in our

manuscript). We have now added the detailed description of these results in Supplementary Note 4 and refer to it in the main text (lines 262-267).

A very essential component of this paper is how the genetic data were processed and analyzed to get to the population-size-through-time estimates. The methods description on this essential and complex element of this study is currently too short (lines 380-411) and lacks important information, making it difficult to impossible to evaluate how well the results are supported by the data. I could not find any more detailed description of these methods in the supplementary material (apologies in case I missed something). See e.g. <https://doi.org/10.1093/molbev/msu058> for an example of a similar methodology described in appropriate detail.

Answer: We now provide additional detail and justification for the steps of the implemented bioinformatic pipeline. The additional text is located in the “Results” section (lines 80-89) “Data curation” section (lines 432-445), “Mapping of short read data” section (lines 475-497) and “Demography inference” section (lines 504-519). Hopefully, this makes the methodology clearer.

Some specific points that definitely need more information are the following:

- line 383: what is the purpose of creating "unmapped bam files with marked adapter sequences"? Was this to remove any remaining Illumina adapter contamination? The purpose is not clear, particularly since in the next step the reads are mapped to references, creating mapped bam files.

Answer: Correct, this step was necessary to remove the Illumina adapters from short reads. Specifically, for creating the mapped BAM files needed as input for the PSMC program, we used the bioinformatic tools which are part of the Genome Analysis Toolkit (GATK; McKenna et al. 2010; 10.1101/gr.107524.110) workflows. We therefore followed the GATK best practices workflow for data pre-processing (<https://gatk.broadinstitute.org/hc/en-us/articles/360035535912-Data-pre-processing-for-variant-discovery>), which requires the conversion of FASTQ files into unmapped BAM (uBAM) files prior to read mapping. The uBAM format is the required input for the next step of the bioinformatic workflow - the MarkIlluminaAdapters program - which is used to mark and remove Illumina adapters from short reads, to avoid potential adapter-related bias when creating the mapped BAM files. We now make this clear in the “Methods” section (lines 478-482).

- line 384: what were the reference sequences used for read mapping? This is very relevant information! The next sentence seems to imply that they were mapped against contigs of a certain size, but it is not clear where those contigs are coming from. Did they come from the same sample via de novo assembly or from some reference taxon that was used? If it is the former, that step needs to be described (how were the contigs generated). After reading through the methods again I found some of the relevant information in the lines 351-355. This text should be moved to go together with the rest of the sequence data methods descriptions, otherwise it's hard to connect the dots.

Answer: We now provide additional information about the reference sequences used for mapping in lines 432-438 and 483-489, as well as refer to Supplementary Table 1, which contains the relevant information (i.e. accessions of both fastq files and reference sequences used for mapping, as well as the source database for each reference). We also now emphasize that we only used previously published reference sequences for mapping. Hopefully, this alleviates the issue.

- can you provide any information on the nature of the genetic loci that were used here? were they always the same for all species, are they just randomly selected from the genomic data and are there any assumptions that can be made about them being neutral markers or under selection? These dynamics matter for effective population size estimation and should we touched upon (can be in the supplementary material).

Answer: To infer the population size trajectory for a species, we follow the common practice outlined in the PSMC recommendations (Li and Durbin, 2011; <https://doi.org/10.1038/nature10231>) and use all available (and therefore random) genomic sites for inference, which in most cases are >80% of the total genome size. We do not have exact information about the percentage of overlap of these sites between species, as the variants were called using species-specific references. To provide an accurate estimate of this overlap, a whole-genome, site-by-site alignment between all reference sequences would be necessary, which is currently unavailable and would require considerable effort to accomplish. We can however say with some confidence that, given the generally high level of synteny between groups of mammalian genomes, sites used for PSMC inference would likely overlap between species (albeit, to an unknown extent).

Additionally, given the fact that approximately 8% of the human genome has been inferred to be evolutionary constrained ([10.1371/journal.pgen.1004525](https://doi.org/10.1371/journal.pgen.1004525)), which is also likely to hold true for other mammals, most sites used are likely to be neutral. Furthermore, if a genomic variant is deleterious it would likely be selected out from the population by purifying selection; whereas, highly beneficial variants would be fixed in the population due to positive selection, and therefore not segregating as variant sites. Furthermore, positive selection tends to be localized in the genome, while we use sites that are distributed genome-wide. Due to these considerations, the majority of observed segregating sites would likely be neutral (or nearly neutral) and so we chose to use all sites for our analysis. We now added discussion about this in the manuscript (lines 516-519).

It should be emphasized more in the main manuscript how the population size through time curves were dated (via estimates of mutation rates for individual species and the detected number of mutations/heterogenous sites in the genetic data). This is key to understanding this study and should be made clear. Supplementary text 1 covers this part in some more detail, but this information deserves a place in the main text of the manuscript. Since the authors are using genome-wide genetic data, I wonder how a single mutation rate can account for the vast variation of mutation and substitution rates present across the genome. There are vast differences in these rates across different loci. It is obviously not feasible/possible to find or produce rate estimates for every genetic locus represented in the data, but this limitation should be highlighted and

discussed. In the current version I could not find any such discussion in the main text or the supplementary text.

Answer: We agree - to provide clarity, we now provide details about the methodology behind the inference of population size trajectories using the PSMC method in the first paragraph of the "Results" section (lines 80-89).

Concerning variable mutation rates across the genome and their effect on PSMC inference - the authors of the original PSMC publication (Li and Durbin, 2011; <https://doi.org/10.1038/nature10231>) have tested the robustness of the method using simulations that include variable genome-wide mutation rates and have found the method to perform reasonably well (Fig. 2b in Li and Durbin, 2011). In humans, mutation rates at the 100 kb scale vary less than a factor of two (Jonsson et al, 2017; [10.1038/nature24018](https://doi.org/10.1038/nature24018)). However, as pointed out by the reviewer, the genome-wide distribution of mutation rates would currently be impossible to obtain for every species. Consequently, using the average mutation rate is presently the only option available to us.

I wonder about the use of the terminology "mutation rates". Wouldn't substitution rates be the more appropriate term here, since the authors are looking at the number of fixed nucleotide polymorphisms (substitutions)?

Answer: Please note that PSMC inference relies on the distribution of *polymorphic* (i.e. non-fixed or heterozygous) sites across the genome - even though we only use sequence data from a single individual, due to the diploidy of the genome, polymorphic sites are observed when the data is mapped to a reference sequence. To clarify further, these polymorphic sites are *not* sites that differ between the sequence data of the individual and the reference sequence, but are sites that have been found to be heterozygous in the individual itself. Therefore, in order to scale the PSMC output into effective population sizes and time (in generations), it is necessary to have knowledge of the mutation (rather than substitution) rates by which the observed polymorphic sites in the genome of the individual originated.

Lines 69-71: The authors refer to the identified "lack of compensatory dynamics". Given that only species above 22kg were considered as megafauna species in this study, would it be conceivable that medium-sized or smaller species that occupy overlapping ecological niches with the disappearing megafauna increased in population sizes and may have displayed some kind of compensatory dynamics? Just because these can't be detected in the megafauna species, it does not mean that such ecological compensation did not happen at all. This should at least be acknowledged or discussed.

Answer: We agree and please note that this has already been acknowledged in the discussion (lines 367-370).

Line 18: "similar declines": I suggest to be more precise in the wording here, because "similar" would mean going extinct.

Answer: We agree. We have thus rewritten the sentence.

Fig 1: The giraffe image has a copyright-watermark on it. Check carefully the copyright license for all these pictures and cite the source where appropriate. I suggest to turn the x-axis in panel A into real numbers (1,000, 10,000, 100,000), to increase readability.

Answer: Corrected. We now use only images with the CC BY-NC copyright and list the sources and authors of the images in Supplementary Table 2. We have also now changed the x-axis.

Lines 75-81: Reading this paragraph makes me immediately curious how the dating of the population dynamics was done. I think that is relevant information for the understanding of these results and should be mentioned here (can be a single sentence, briefly mentioning the dating method).

Answer: We agree and have now provided more details in this paragraph that explain the PSMC methodology (lines 80-89).

Line 304: Remove the ", indicating"

Answer: Corrected.

Reviewer #2 (Remarks to the Author):

In this report, Bergman et al. concisely and clearly report estimates of historical population sizes for a diverse group of megafauna, demonstrating a link between this decline and human dispersal. The piece is superbly written and the science rigorously executed.

While the article is excellent, I would like the authors to reflect on the possible effects of population structure, multiple testing, and phylogeny on their inferences. It would be interesting to discuss the extent to which increasing population structure through time - associated with population declines - might have impacted the PSMC results.

Answer:

Thank you for your positive assessment!

Concerning population structure: We have now added discussion about the effect of population structure on PSMC results (lines 352-362). While we acknowledge that population subdivision and migration patterns can create patterns of increase and decrease in PSMC trajectories, there is currently no reliable methodology available to distinguish between population structure in subdivided populations and population size change in a panmictic population. While episodes of differential population subdivision and connectivity could theoretically be fitted to explain the population trajectory for each species using simulation, this would require considerable additional effort, as well as an additional justification for the inferred historical population structure of each species. We also note that it is highly unlikely that population structuring for all of the 139 studied megafauna species (which differ with respect to their life history traits, habitat preference and

geographic distributions) would have been so similar as to manifest as a global decline with an onset during a very narrow time window, which largely corresponds to global *Homo sapiens* expansions. Therefore, we consider *Homo sapiens* expansions to be a more parsimonious explanation to megafauna declines, rather than changes in population structure.

Concerning multiple testing: As we conduct our analyses in a Bayesian framework, this means that the classical multiple testing consideration has a different (and not completely equivalent) treatment compared to a classical frequentist statistical approach. With Bayesian modeling, we need to explicitly assume a prior distribution for a parameter of interest, which is then updated given data, and finally translated into a posterior distribution - due to the explicit need to define priors, the Bayesian approach is conservative and the need for multiple testing is alleviated. In other words, each hypothesis is evaluated independently based on its own posterior probability, without being directly influenced by the number of tests performed. Additionally, we model the species-specific relationships between variables in a partially pooled manner, where we assume a cross-species pooled error distribution for every model and sometimes a pooled distribution of species-specific intercepts and slopes, as well assume weak priors (i.e. usually centered at 0 if we consider slopes of relationships), which could be considered as accounting for multiple testing within the Bayesian framework. These considerations make it such that the need for explicit correction of multiple testing in Bayesian modeling is largely unnecessary. We have added a comment about this in Supplementary Note 1 (lines 86-89).

Concerning phylogeny: Closely related species will share their population size history during the period preceding speciation, which may affect the observed population dynamics. To test this claim more rigorously, we subset our dataset to 67 species (i.e. one representative species per studied genus) in order to focus on a subset of species that are most likely to *not* share evolutionary history over the last 740,000 years (for which we conduct the majority of our analysis). We then repeated the analyses from Fig. 1 using this species subset and found that the results remained largely unchanged. We have also conducted a phylogenetic regression analysis (as per reviewer's next suggestion) and again found little explanatory power of phylogeny on our results. We therefore conclude that the majority of the studied species have likely become distinct evolutionary units over the last 740,000 years) and shared evolutionary history likely plays a minor role in the observed patterns of megafauna dynamics. These results are now described in Supplementary Note 3 and main text lines 141-154.

Similarly, the authors might want to comment or directly address the impact of phylogeny versus body size on their results. This could be done via phylogenetic regression. Concretely, if all large animals are also very closely related, it is difficult to point to body mass over other inherited traits that might increase their chances of population decline. Similarly, body mass is associated with a broad range of life history traits, so it might be worthwhile to at least speculate on what might be the root cause of the susceptibility of large species to population decline in the hands of early human societies. For instance, small clutch sizes might be the ultimate cause of a slow replenishing of populations.

Answer: Mass was chosen as a trait of interest primarily because of the fact that recent extinctions have been highly skewed towards species with large mass (Smith et al, 2018; 10.1126/science.aao598) - we therefore wanted to test whether declines in extant megafauna also exhibit this correlation. Additionally, we note that the majority of relevant life history traits are correlated to mass, and it would thus be difficult to disentangle the effect of mass from other factors as well as point to causative effects. We have however conducted a phylogenetic regression analysis and found that the relationship between the severity of decline and species mass remained largely unchanged when controlling for phylogenetic relatedness. This analysis is described in Supplementary Note 3.

The authors might have also addressed already the repeated tests using of temperature data across species. It would be worth making sure that such a correction is implemented and/or described in the text.

Answer: We now provide more justification about the consideration of multiple testing in a Bayesian framework in Supplementary Note 1 (lines 86-89).

Reviewer #3 (Remarks to the Author):

Noteworthy results:

The most noteworthy results of the analyses presented are a that there is a significant relationship between the timing of human arrivals across continents and the extinction of megafauna, but no significant relationship with climate. Additionally this work presents estimates for megafauna census sizes, biomass and energy turnover between the Last Interglacial and the current period. Overall these estimates make for a compelling argument of the severe ecological impacts of end Pleistocene megafauna extinctions.

Significance to the field and related fields:

I believe that this work has the potential to be significant to fields that take interest in late Pleistocene extinctions as well as modern conservation. It does not present an overall result that is new, the consequences of the loss of megafauna is widely discussed in this field and the debate over human vs. environmental causes is many decades old. However, this paper offers quantification of the consequences of megafauna extinction that are striking and the overall approach, with some caveats, is sound.

Support for the conclusions and claims

The work supports the conclusions and claims, however there are concerns with how the authors have used some previously published data from other investigators for their purposes.

Flaws in the data analysis, interpretation and conclusions that require revision:

My most significant critique of this study is the use of previously published datasets beyond what is appropriate and the resulting overinterpretation of these data. While I am inclined to agree with the authors that climate is probably not playing the primary role in continental/global scale patterns

of extinction, this paper severely overstates what they have uncovered about climate's role in driving effective population sizes, or lack thereof. There is an abundance of literature documenting AET, min/max temperature, or CV of temp (among others) as better predictors of diversity and distribution of species than even regional MAT. These aspects of climate don't necessarily tightly covary with MAT, so MAT is not a catch all or stand in for "climate." What the authors have demonstrated is that global MAT is not a driver, and they can confidently state that. However extrapolating their findings to all aspects of climate, which they appear to be doing, is a reach.

Response: Thank you for this comment - in the hope to alleviate this issue, we have now relaxed the language surrounding the description of the climate effect and have also added annual precipitation as an explanatory variable in our models; consequently, we have now conducted additional modeling of the climate effect on megafauna dynamics (new Supplementary Table 12). We have found that including precipitation in the models of megafauna dynamics did improve model fit - however, we also found that the overall patterns and conclusions remained unchanged. Namely, we still observed an increasingly worse fit of climate-only models for recent time periods (new Fig. 2b), as well as a significantly higher explanatory power of models that included human arrival time as predictor (new Fig. 3a).

Furthermore, there needs to be some justification provided for the extrapolation of MAT taken from a singular Antarctic ice core to the entirety of the globe. I find this concerning because we know that there can be large differences in the records documented between ice cores and regional proxy records that are more appropriate for the "realms" being investigated in this study. For examples of the variation I am referring to see Fiedel 2011 in Quaternary International and Fastovich et al. 2020 in Geophysical Research Letters. This problem is especially pertinent to lines 325-327 in the discussion.

Answer: We now ensure that it is more consistently acknowledged that we use mean annual temperature (and precipitation) as proxy for climate at several points in the manuscript (lines 170-174, 221, 258, 387).

The main justification for why temperature was used as a proxy for climate is that temperature is the most frequently studied parameter of paleoclimatic conditions and is widely used as a general climatic indicator (alongside annual precipitation), with multiple sources of comparison that produce generally similar estimates. We therefore opted to use the Antarctic ice core as a representative of this parameter. Furthermore, as the size of the time windows for which we calculate average population sizes and temperature values in our analyses is large (for example, 25,000-year windows over the last 100,000 years and even larger time windows for more ancient time periods), we argue that global temperature (and precipitation) dynamics are an acceptable determinant of population dynamics at these large time scales, where the global climate system has been dominated by glacial-interglacial oscillations. However, we have added an acknowledgement that more fine-scale, abrupt climate dynamics, such as the 1300-year-long Younger Dryas Younger period referred to by the reviewer, as well as climatic parameters other than temperature, could have played a role in determining population dynamics. Discussion on these points is not added in lines 382-394.

To further address the comments by the reviewer, we have now conducted additional analyses using annual precipitation as a complementary proxy for climate. We have tested multiple additional models with different combinations and effects of the climate variables (new Supplementary Table 12). While the overall fit of the climate-only model improved, the conclusions of the initial submission remained unchanged.

Soundness of methodology:

The approach is logically sound and is clever, however I stand by my critique of their use of previously published data.

Response: While it is true that no new data has been generated for this study, we paid particular attention to using only the highest-quality genomic datasets, as well as applying rigorous filtering procedures (described in lines 442-445, 475-497, 500-519) to estimate megafauna population trajectories to the highest possible accuracy. We also note that genomic data provides an unprecedented level of detail when it comes to assessing past population sizes of species, compared to other sources such as the fossil record, thus making it ideal to study determinants of past population dynamics. In the new version of the manuscript, we have also added an additional climatic variable - annual precipitation - and tested more complex models of the climate effect on megafauna population dynamics in the hope to provide a better representation of the climate effect.

Reproducibility of the analyses:

This paper uses an approach that is purely mathematical and there appears to be enough detail provided in the methods for the work to be reproduced. It would be of value for future work to use their approach with alternative datasets that lack the issues I have laid out.

Other specific comments:

Line 17: This dichotomy is increasingly seen as a strawman argument, there are a growing number of paleoecologists who hold a more nuanced perspective that both human and climate have varying degrees of impacts on extinct megafauna at different scales

Answer: We agree, and have now rewritten this sentence.

Figure 1: Some of the animal icons in figure 1 have clear copyright watermarks on them, please ensure that appropriate rights are obtained for images and figures.

Answer: Corrected. We now use only images with the CC BY-NC copyright and list the sources and authors of the images in Supplementary Table 2.

Line 139: The particulars of how climate plays out on landscapes, especially in the interior of continents, can vary considerably from what is documented in ice core data. In other words, it may not reflect the climate conditions that are experienced by organisms.

Answer: While we generally agree with this statement and now acknowledge it in the manuscript (lines 382-389), given the nature of our data - i.e. large time windows over which we calculate average population sizes and temperatures, as well as the fact that we study species with generally large geographic ranges - we argue for the validity of the conducted analysis. We also note that the dominant climate dynamics within our time frame are glacial-interglacial oscillations, which indeed are a global phenomenon affecting all areas to a substantial, albeit not identical extent. Ice cores have played a central role in the study of these dynamics. Additionally, while complex interactions between regional or local climate and landscapes is likely to have affected species, the appropriate datasets needed to test this interaction over the inferred time windows of the population size trajectories are not available. However, given the high consistency in the patterns documented by our analysis, any such complexities have not been strong enough to obscure the general trend.

Lines 158 - 159: Global temperature records are widely available, making them quite convenient, but we know that ecosystems respond to different aspects of the climate beyond mean annual temperature. In many cases precipitation is as influential, or more so, than temperature. For this reason, it's not all that surprising that a significant pattern isn't observed with MAT.

Answer: We note that MAT is broadly used as a general climatic indicator and captures the overall glacial-interglacial climate oscillations, the major climatic dynamic of the study period, very well. However, we agree that precipitation also has potential to be influential and have now conducted additional analysis that used precipitation (as stand-alone or in combination with temperature and/or human predictors) as an explanatory variable (list of tested models that include precipitation is provided in the new Supplementary Table 12). We again found no significant explanatory effect of precipitation on megafauna dynamics.

Lines 236-239: Data that has been log-normalized has variance that is compressed relative to the raw values and yet the strength of this correlation is in the mid-range of what might be considered as only "moderate." This should be acknowledged, and I would like to see the authors clearly explain how this impacts the calculated values of total megafauna census size, biomass, and energy turnover.

Answer: We thank the reviewer for this comment which prompted us to revise the analysis of the consequences of megafauna decline (Fig. 4). Specifically, we now use a simpler model to infer historical megafauna census sizes, that is based on the ratio of current (Holocene) effective to census (IUCN-estimated) size using un-transformed population size measures (lines 274-278 and Supplementary Note 1). This new approach gave us somewhat higher estimates of the current reduction of megafauna census sizes compared to the original analysis. Therefore, the original approach based on the N_e - N_c relationship in the log-normalized form likely produced, on average, underestimates of census size reductions. Notably however, both analyses are likely to result in underestimations of the current reduction of megafauna census sizes, as they are both based on estimates of current census sizes of species that underwent population contractions in the recent past. Therefore, in the Discussion section, we now present an additional analysis which attempts to correct for this underestimation (lines 402-419).

We have still kept the original estimates of total megafauna census size, biomass and energy turnover predicted from the log-log relationship of two population size measures in Supplementary Notes 5 and 6 - we believe they are still a useful representation of the lower bound of current megafauna census size reduction. Additionally, this approach also has the advantage to produce distributions of the three parameters, rather than point estimates. Furthermore, as the estimates of the initial approach vary over orders of magnitude (Supplementary Figs. 18, 21 and 22), the estimated distributions likely capture the majority of the actual range for the parameters. Albeit, the large variance of these distributions is likely a consequence of the moderate correlation strength between the two population size measures, which we have now acknowledged in Supplementary Note 5 (lines 136-148).

Line 304: ", indicating." this looks like a typographical error.

Answer: Corrected.

Lines 344-346: How similar IS the current interglacial to the last interglacial? Such a claim needs references.

Answer: We have now added two references for this claim (line 418; [10.1098/rsta.2013.0097](https://doi.org/10.1098/rsta.2013.0097), [10.1126/science.aai8464](https://doi.org/10.1126/science.aai8464)).

Line 353: How was it determined if a sample was representative? Ne can vary greatly regionally and across populations, how is this addressed?

Answer: First, please note that our analyses are based on mammal species and associated samples from broadly around the globe, so in terms of overall patterns there is little reason to doubt our sample is reasonably representative in this regard. The limitation of data availability was the main determinant of a representative sample at more detailed levels. While it would be ideal to study regional differences between different populations of a single species, for the majority of the studied species data of only a single population was available to infer the PSMC trajectory. The short-read data needed to produce the PSMC trajectory was usually, but not always, available from the same individual used for the assembly of the reference sequence. When the short-read data was not available for the reference individual, we searched the databases for an individual of the same species. We also searched the databases for short-read data of individuals that are representative of different populations (or subspecies) within the species complex and included them in our analysis. When data of multiple individuals were available for a single species, we chose the individual with the largest amount of short-read data as representative, such that we maximize genomic coverage of the mapped bam file and thus ensure higher accuracy of the PSMC estimates. We now make this clear in the Methods section (lines 441-445). We also note that the coalescent history of an individual over hundreds of thousands of years will likely represent the whole species rather than a single population unless populations have been completely separated over an extended period of time.

Reviewers' Comments:

Reviewer #1:

Remarks to the Author:

Thank you for this thorough revision of my previous comments and for this great scientific contribution to the field of macroecology. Please check my few remaining comments below, mostly regarding the effective human population size trajectory, this part requires some more attention. Other than those remaining issues, I am happy with the revisions and the state of the manuscript.

Additional comments:

Thank you for going through the effort of adding the human effective population size modeling to this manuscript, I believe this is an important addition. I don't understand, however, why this was done on simulated and not empirical sequence data for *Homo sapiens*. It seems to be circular reasoning to simulate sequence data based on a model of human demography and then estimate the effective population size from those simulated data. The described methodology by Chen et al (2019) seems to be based on the original publication by Li & Durbin (2011, 10.1038/nature10231), where they used simulated genetic data only to validate the PSMC method, but then they analyzed empirical data with the method (see Fig. 3 in their article).

The thought behind my original comment was that this could provide an additional and independent insight of what the human effective population size trajectory looks like if the methods described in this manuscript were applied to real genetic data of *Homo sapiens*. Does it increase in the timeframe of this study, contrary to the patterns of all megafauna species (expected)? This would help to judge whether the PSMC method can reliably detect the approximate trajectory of population sizes during this time frame, using genomic data. I therefore suggest the authors apply the PSMC method to real empirical (!) human genomic data, using the same workflow as was used for all other species.

A sidenote: This additional analysis is described in Supplementary Note 4, which refers to Supplementary Figure 8. Please note that the results are not shown in Fig S8, but in Fig S14 instead (make sure to update the numbering of the supplementary figures in any text where they may be referenced).

Lines 89-92: "In total, we inferred population dynamics from 139 megafauna species genomes (Fig. 1a) and observed a general decreasing trend towards present time, as demonstrated by a negative correlation between effective population size and time before present."

 The way this is phrased, I would expect that a negative correlation between effective population size and time before present would in fact mean a population increase. As we move towards the present the time-before-present values get smaller, and, if the correlation is negative, the population size values would have to increase towards the present. I suspect this is just a wording issue and that the time-axis was in fact scaled in reverse? Please make sure to fix this in the description or in the correlation analyses.

Best,

Tobias Andermann (reviewer 1)

Reviewer #2:

Remarks to the Author:

The authors have made a remarkable job in addressing the reviewers' comments. This is satisfactory and makes the article an outstanding contribution to the field of recent megafaunal population dynamics.

Reviewer #3:

Remarks to the Author:

The authors have addressed the main issues I noted during my initial review of the manuscript. Most notably, they have tempered their discussion/conclusions to match what is discernible from the results produced by the data they have. I am satisfied with the additional analyses they conducted. I also find the additional explanations and justifications they provide, in discussing the results of their modeling and estimates of megafaunal impacts, to be compelling.

The authors have also addressed and corrected some small typos I noted in my initial review.

Reviewer #4:

Remarks to the Author:

In this manuscript, the authors report estimates of population size trends across Eras and biogeographic realms, compare them to both human population trends and climatic parameters, demonstrating a link between human presence and decline in megafauna. Working across realms and with a multi-species perspective adds relevance and support to the findings. The manuscript is clearly and concisely presented, and the science rigorous. The authors appropriately and convincingly addressed all comments from a previous round of review, and I believe the manuscript is now ready for publication, with just some minor edits.

General comments:

- Numbering of tables, figures and supplementary should follow the order of first mention in the main text. Reference style to table and figures should also be consistent across all the files.
- Please check with a native speaker on the use of "suppressed" in the text. I am not a native speaker myself, but on a few occasions it did not sound like the most appropriate word choice.

Specific comments:

L51: typo in modelling

L239: the text says "species-specific", whereas the legend says "across species", please correct as appropriate. If species-specific, please mention which species was chosen.

L458-460: the subject is "Geographic classification", so the verb should be "is".

Supplementary Note 3, L8: observed is repeated twice.

Supplementary Note 3, L9-10: please specify the threshold of ancient/recent.

RESPONSE TO REVIEWERS

Reviewer #1 (Remarks to the Author):

Thank you for this thorough revision of my previous comments and for this great scientific contribution to the field of macroecology. Please check my few remaining comments below, mostly regarding the effective human population size trajectory, this part requires some more attention. Other than those remaining issues, I am happy with the revisions and the state of the manuscript.

Additional comments:

Thank you for going through the effort of adding the human effective population size modeling to this manuscript, I believe this is an important addition. I don't understand, however, why this was done on simulated and not empirical sequence data for *Homo sapiens*. It seems to be circular reasoning to simulate sequence data based on a model of human demography and then estimate the effective population size from those simulated data. The described methodology by Chen et al (2019) seems to be based on the original publication by Li & Durbin (2011, 10.1038/nature10231), where they used simulated genetic data only to validate the PSMC method, but then they analyzed empirical data with the method (see Fig. 3 in their article).

The thought behind my original comment was that this could provide an additional and independent insight of what the human effective population size trajectory looks like if the methods described in this manuscript were applied to real genetic data of *Homo sapiens*. Does it increase in the timeframe of this study, contrary to the patterns of all megafauna species (expected)? This would help to judge whether the PSMC method can reliably detect the approximate trajectory of population sizes during this time frame, using genomic data. I therefore suggest the authors apply the PSMC method to real empirical (!) human genomic data, using the same workflow as was used for all other species.

Response: With this analysis, we wanted to capture the correlation between *global* human and megafauna population dynamics. The reason we simulated a human genome was to create an individual that encompasses genetic variation from different human populations, and therefore hopefully also captures the dynamics of the global effective population size of humans. When using empirical human data, the choice of an individual from a single population would not reflect global, but rather (more or less) localized population size dynamics. The geographic localization of human populations is especially pertinent for recent time points as we know from archeological and genomic data that humans have had an increasingly complex population history towards present time, especially after the onset of genetic diversification of different human populations, followed by Out-of-Africa migrations which are characterized by continuous bottleneck and founder events across the globe (Bergstrom et al., 2021; 10.1038/s41586-021-03244-5).

While the authors of Li and Durbin (2011) indeed use simulated data to validate the PSMC method, the authors in Chen et al. (2019) use the trajectory inferred from the simulated African-Asian genome to demonstrate the global dynamics of human effective population size against global dynamics of ruminant population sizes (Figure 2A in their main text; please note that their x-axis is reversed in relation to our figures) - we therefore initially chose to follow their example as we believe it better reflects global, and not just local change of human effective population size, and therefore provides a meaningful comparison to global megafauna population dynamics.

We have also now estimated population dynamics of an African and Asian individual from empirical data using the PSMC method. Specifically, we found that PSMC trajectories between 1,500,000 and 150,000 ya correspond well between all three individuals (African, Asian and the simulated African-Asian genome), as well as correlate negatively with global megafauna dynamics. However, between 150,000 and 50,000 ya both the African and Asian populations show signs of bottleneck dynamics, likely due to structuring and migration of the human population (Bergstrom et al., 2021; 10.1038/s41586-021-03244-5). Consequently, when correlating the full trajectories (between 1,500,000 ya and the present) of megafauna and human individuals, only the trajectory inferred from the simulated African-Asian genome still correlates negatively with the global megafauna trajectory, while the correlation becomes non-significant for the trajectories of the African and Asian individual. However, the population size trajectory inferred from both the African and Asian genomes start increasing between approximately 50,000 ya and present time, in stark opposition to the megafauna trend, which experienced their lowest population sizes during this time frame (new Supplementary Fig. 14). We however caution that due to increasing geographic localization of human populations close to present time, it becomes questionable to relate recent human population dynamics inferred from specific individuals (and populations) to the global (or even realm-specific) megafauna trend.

We now describe these new results in the main text (lines 265-271), new Supplementary Note 4 and Supplementary Fig. 14. We thank the reviewer for the comment and hope that this additional analysis addressed the remaining concern.

A sidenote: This additional analysis is described in Supplementary Note 4, which refers to Supplementary Figure 8. Please note that the results are not shown in Fig S8, but in Fig S14 instead (make sure to update the numbering of the supplementary figures in any text where they may be referenced).

Response: Thank you for bringing our attention to this - we have now made the correction!

Lines 89-92: "In total, we inferred population dynamics from 139 megafauna species genomes (Fig. 1a) and observed a general decreasing trend towards present time, as demonstrated by a negative correlation between effective population size and time before present."

 The way this is phrased, I would expect that a negative correlation between effective population size and time before present would in fact mean a population increase. As we move

towards the present the time-before-present values get smaller, and, if the correlation is negative, the population size values would have to increase towards the present. I suspect this is just a wording issue and that the time-axis was in fact scaled in reverse? Please make sure to fix this in the description or in the correlation analyses.

Response: We apologize for this oversight. You are correct - population size decreases towards the present and should therefore increase (and correlate positively) with increasing time before present. We have now corrected this sentence and the correlation coefficient (lines 91-92).

Best,
Tobias Andermann (reviewer 1)

Reviewer #2 (Remarks to the Author):

The authors have made a remarkable job in addressing the reviewers' comments. This is satisfactory and makes the article an outstanding contribution to the field of recent megafaunal population dynamics.

Response: Thank you for your comments and we are happy that you found our revision satisfactory.

Reviewer #3 (Remarks to the Author):

The authors have addressed the main issues I noted during my initial review of the manuscript. Most notably, they have tempered their discussion/conclusions to match what is discernible from the results produced by the data they have. I am satisfied with the additional analyses they conducted. I also find the additional explanations and justifications they provide, in discussing the results of their modeling and estimates of megafaunal impacts, to be compelling.

The authors have also addressed and corrected some small typos I noted in my initial review.

Response: Thank you for your comments and we are happy that you found our revision satisfactory.

Reviewer #4 (Remarks to the Author):

In this manuscript, the authors report estimates of population size trends across Eras and biogeographic realms, compare them to both human population trends and climatic parameters, demonstrating a link between human presence and decline in megafauna. Working across realms and with a multi-species perspective adds relevance and support to the findings. The manuscript is clearly and concisely presented, and the science rigorous. The authors appropriately and convincingly addressed all comments from a previous round of review, and I believe the manuscript is now ready for publication, with just some minor edits.

Response: Thank you for your comments and we are happy that you found our manuscript satisfactory.

General comments:

- Numbering of tables, figures and supplementary should follow the order of first mention in the main text. Reference style to table and figures should also be consistent across all the files.

Response: We have now made an effort to correct this; thank you for the comment.

- Please check with a native speaker on the use of “suppressed” in the text. I am not a native speaker myself, but on a few occasions it did not sound like the most appropriate word choice.

Response: We agree, and have changed the wording of one of the sentences (line 425).

Specific comments:

L51: typo in modelling

Response: Corrected.

L239: the text says “species-specific”, whereas the legend says “across species”, please correct as appropriate. If species-specific, please mention which species was chosen.

Response: We have corrected this (lines 239-240).

L458-460: the subject is “Geographic classification”, so the verb should be “is”.

Response: Corrected.

Supplementary Note 3, L8: observed is repeated twice.

Response: Corrected.

Supplementary Note 3, L9-10: please specify the threshold of ancient/recent.

Response: We now added this information.

Reviewers' Comments:

Reviewer #1:

Remarks to the Author:

Thank you for addressing my remaining comments and for including the additional analyses. I have no more comments and can fully endorse the manuscript for publication.

Congratulations to an interesting and comprehensive study and manuscript!

Reviewer #4:

Remarks to the Author:

The authors have sufficiently addressed all comments from the previous round and the manuscript is now ready for publication.